# Boosting Visual-Language Models by Exploiting Hard Pairs

## Abstract

Contrastive Language-Image Pre-training (CLIP) has emerged as the industry standard for aligning images with their corresponding textual descriptions. However, to enhance zero-shot recognition, current methods often demand additional data collection and retraining with the introduced new loss functions. Such additional demands for data collection and retraining impose substantial constraints on their practical deployment. In this work, we present HELIP, a low-cost strategy tailored to enhance the performance of pre-trained CLIP models. This is achieved by further training them with challenging text-image pairs selected from their training dataset. Our proposed Hard Pair Mining (HPM) method treats a text-image pair as a single point in the joint Vision-Language space and identifies those in close proximity to a given pair as its hard pairs. By incorporating these challenging data, we refine pretrained CLIP models using both the traditional contrastive alignment loss and the newly introduced Hard Negative Margin Loss (HNML). This approach ensures the optimal harnessing of insights from challenging data. Notably, HELIP is designed to be seamlessly integrated with existing models, providing an enhancement without the need for training a model from scratch or collecting additional data. On a comprehensive zero-shot and retrieval benchmark, HELIP consistently boosts existing models to achieve leading performance. In particular, for ImageNet zero-shot accuracy, HELIP boosts CC3M and CC12M pretrained SLIP by 3.05 and 4.47 respectively. In addition, the systematic evaluations of zero-shot and linear probing experiments across fine-grained classification datasets demonstrate a consistent performance improvement and validates the efficacy of HELIP. Specifically, HELIP boosts the zero-shot performance of pretrained CLIP and SLIP by an average of 8.4% and 18.6%, respectively, and improves their linear probe performance by an average of 9.5% and 3.0%.

## 1 Introduction

Contrastive Language-Image Pretraining (CLIP) (Radford et al., 2021) is quickly becoming the standard for foundation models in computer vision due to its effectiveness for a variety of vision-language tasks without task-specific finetuning (i.e. zero-shot), such as image classification (Li et al., 2021) and image-text retrieval (Baldrati et al., 2022). Nevertheless, web-crawled image-text pairs used for the training of CLIP models are often loosely connected, leading to multiple plausible matches, beyond the assigned ones (Wu et al., 2022). Hence, a number of strategies (Li et al., 2022a; 2021; Mu et al., 2022; Radenovic et al., 2023) have been presented to investigate appropriate matches and take advantage of the widespread supervision among the image-text pairs to improve language-image pretraining.

Efforts to improve the foundational contrastive language-image pretraining have largely followed two distinct approaches: 1) the incorporation of a multitask objective to bolster the efficiency of single-modality monitoring (Li et al., 2022a; Mu et al., 2022); and 2) the use of intra/inter-modality similarities to identify and retrain with sample-level challenging data (Li et al., 2021; Radenovic et al., 2023). Notably, techniques that depend on intra/inter-modality similarities for intensive negative data mining within a batch frequently fail to pinpoint data beneficial for contrastive learning. This isn't solely due to batch size constraints, but also because the contrastive loss in CLIP is over pair data; thus, sample-level hard data may not always be effective. This issue is further compounded when imprecise matching of image-text caption pairs leads to inaccurate

hard pairs. Given these issues, a natural question arises: **Can the performance of pre-trained CLIP models be improved more efficiently and broadly using hard data from its original pretraining dataset?**

In response to the question, we introduce the innovative training framework HELIP as a tool to enhance CLIP models. This framework boosts CLIP models by effectively using hard pairs derived from the original training sets. Contrary to traditional methods that select hard samples based on the intra/inter-modality similarity calculated within a batch, HELIP firstly defines hard pairs as nearby pairs within a joint Vision-Language space. Following this, the challenge of mining nearby pairs is transformed into a feasible proxy task focused on maximizing text-image agreement of the target pair. In this proxy task, the target pair is treated as a matching pair, with the optimization objective being to curate a pair set that enables the proxy model to assign a higher probability of matching. To optimize this agreement, the **Hard Pair Mining (HPM)** component of HELIP is designed to explicitly represent the target text-image pair using the remaining dataset. Subsequently, it then selects a subset that most effectively identifies the target as a matching pair, designating this as its set of hard pairs, as visualized in Figure 1. Additionally, rather than further training CLIP models solely with the original text-image contrastive loss (Radford et al., 2021)—which uniformly pushes all negative samples away from their positive counterpart—HELIP integrates the **Hard Negative Margin Loss (HNML)** into the loss function. As illustrated in Figure 2, the intrinsic similarity between pairs should be reflected in the learned representations. Therefore, during training, HELIP imposes additional geometric structure on the learned representation space by involving HNML as a regularization. This process allows hard negatives to situate themselves closer to the positive pair than the ordinary negatives. In doing so, the valuable information embodied within hard data can be harnessed more effectively.

Empirical tests underscore the efficacy of HELIP. When fine-tuning established CLIP models (such as CLIP, SLIP, and DECLIP) with hard examples and the hard negative margin loss, HELIP consistently enhances CLIP checkpoints across zero-shot classification, text-image retrieval, and fine-grained linear probe benchmarks. For zero-shot classification on ImageNet, CIFAR-10, and CIFAR-100, HELIP consistently boosts the performance of 6 pretrained models. Particularly, using HELIP to boost SLIP models pretrained on CC3M, CC12M, and YFCC15M results in ImageNet zero-shot accuracy gains of 3.05, 4.47, and 10.14, respectively. Further, after finetuning with hard pairs and hard negative margin loss, those pretrained models achieve better zero-shot and linear probe performance on 7 fine-grained image classification datasets. Specifically, the average zero-shot accuracy of CC3M pretrained CLIP and SLIP are improved from 14.45 to 15.67 (+8.4%) and from 16.96 to 20.124 (+18.6%). The average linear probe accuracy of CC3M pretrained CLIP and SLIP are improved from 53.29 to 58.34 (+9.5%) and from 64.89 to 66.81 (+3.0%).Additionally, the performance gain is also valid in terms of zero-shot retrieval, with 1.1 of R@1 on Flickr30K, and 2.2 of R@1 on COCO for SLIP-HELIP. Our contributions could be summarized as:

- To our best knowledge, HELIP stands out as the first method aimed at enhancing already well-trained CLIP models with a lost-cost strategy. It accomplishes this by further training them using their original datasets and is distinctively designed for seamless integration with various CLIP methodologies.

- We propose an innovative technique for selecting hard pairs, specifically targeting the identification of challenging negative pairs. Complementing this, we introduce the hard negative margin loss, an approach that considers representation distances, ensuring the successful incorporation of hard pairs during finetuning.

- Through empirical analysis across zero-shot classification, image-text retrieval, and linear probe benchmarks, we demonstrate that HELIP is able to consistently improve CLIP model checkpoints.

## 2 Related work

**Vision-Language pre-training.** Vision Language Pretraining (VLP) is a technique that leverages large-scale image-text datasets to learn a strong joint representation between the two modalities that can be transferred to various downstream vision-language tasks. VLP models can be generally divided into single-stream models and dual-stream models. Dual-stream models (Jia et al., 2021; Li et al., 2022b; Mu

et al., 2022; Radford et al., 2021; Yao et al., 2022) typically consist of two separate encoders for image and text respectively and perform cross-modality interactions on the top, are becoming more and more popular because of its flexibility of transferring pre-trained knowledge to downstream tasks. CLIP (Radford et al., 2021), uses a simple contrastive objective to learn visual features from natural language supervision and achieves remarkable zero-shot recognition performance using 400M web-crawled image-text pairs. Recent works boot the performance of CLIP by applying self-supervision within visual modal (Mu et al., 2022), additional nearest neighbor supervision (Li et al., 2022b). These methods are actually doing data augmentations to increase data efficiency and thus bring additional computational costs.

**Contrastive learning with hard negative samples.** Contrastive learning learns a representation of input data that maps semantically comparable examples close together and semantically dissimilar examples far apart (Chen et al., 2020a;b; Wang & Isola, 2020). Recent works include hard negative samples into the loss function and achieve better empirical performance (Cai et al., 2020; Huynh et al., 2022; Kalantidis et al., 2020; Li et al., 2021; Radenovic et al., 2023; Robinson et al., 2021; Shah et al., 2022). For Language-image contrastive learning, current approaches (Li et al., 2021; Radenovic et al., 2023) mine multimodal hard negative examples using intra/inter-modality similarity. Li et al. (2021) choose in-batch hard negative samples with image-text contrastive loss. Hard negative noise contrastive multimodal alignment loss by Radenovic et al. (Radenovic et al., 2023) up-weights the loss term for in-batch hard samples. For previous intra/inter-modality hard sample mining methods, two text-image pairs are considered as hard samples, if the cosine similarity between visual/textual features is high (Li et al., 2021; Radenovic et al., 2023). However, due to the nature of loose assignment for web-crawled image-caption data, a high similarity indicated by intra/inter-modality doesn't indicate that the two pairs are difficult to tell apart. Contrary to prior works, we design a hard sample mining method to discover similar pairs defined in joint vision-language space and efficiently select samples challenging enough to improve learning.

## 3 Hard pairs for visual-language models

In this section, we first define the notations and revisit CLIP for zero-shot recognition in the preliminary section. Next, we introduce the Hard Pairs Mining method, denoted as **HPM**, along with the associated Hard Negative Margin Loss, **HNML**, specifically designed to leverage hard pairs identified by HPM.

### 3.1 Preliminaries

We consider the task of contrastive image-text pretraining. Given an image-caption dataset $\mathcal{D} = \{z_i\}_{i=1}^{N} = \{(x_i^I, x_i^T)\}_{i=1}^{N}$, $(x_i^I, x_i^T) \in \mathcal{I} \times \mathcal{T}$, the $x_i^I$, $x_i^T$ denote the image and its corresponding caption, $\mathcal{I}$ and $\mathcal{T}$ indicates visual and textual space respectively, and $\mathcal{I} \times \mathcal{T}$ indicates the joint Vision-Language space. Our goal is to learn a dual encoder model $\phi = \{\phi_{image}, \phi_{text}\}$, where $\phi_{image}$ represents the image encoder and $\phi_{text}$ denotes the text encoder. We use the shorthand $I_i = \phi_{image}(x_i^I)$ and $T_i = \phi_{text}(x_i^T)$ to denote the encoded representation of an image and its caption, respectively. The contrastive objective of CLIP is formulated as,

$$\ell_{CLIP} = -\frac{1}{|B|} \sum_{i \in B} \log \frac{\exp\left(sim(I_i, T_i)/\sigma\right)}{\sum_{j \in B} \exp\left(sim(I_i, T_j)/\sigma\right)}, \tag{1}$$

where $sim(\cdot, \cdot)$ is the cosine similarity function, $B$ is a batch of samples and $\sigma$ is a trainable parameter controlling the temperature. Intuitively, the above formulation explicitly aligns the representations of image and text from one pair.

### 3.2 HPM: hard pair mining

In the context of vision-language contrastive learning, we define "hard pairs" as the pairs that are nearby to a specified target pair within the joint vision-language space, $\mathcal{I} \times \mathcal{T}$. The formal definition of the hard pair mining problem can be found in Equation 2. Here, $z_i$ signifies a target pair, $\mathcal{H}_i$ denotes a set of pairs chosen from the dataset $\mathcal{D}_i = \mathcal{D} \setminus z_i$, and the metric $\mathbf{S}(,)$ quantifies the similarity between the target pair and a set

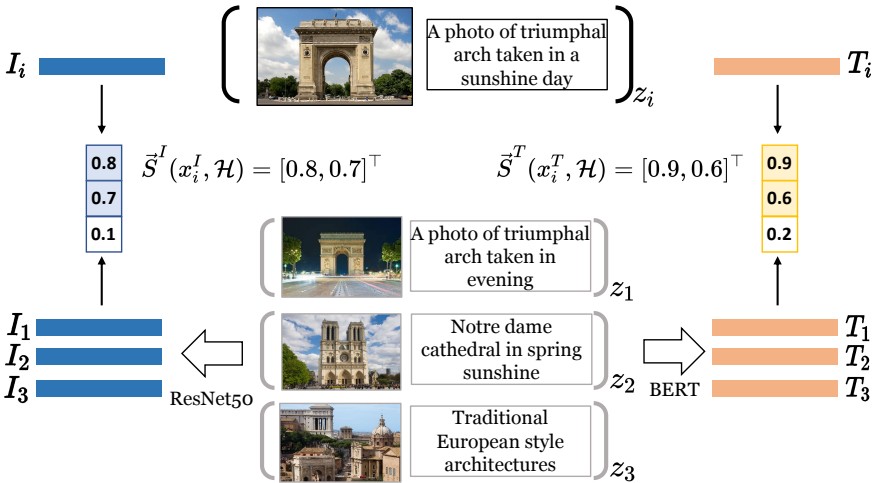

Figure 1: **Hard Pair Mining (HPM)**. Choose hard pairs by optimizing the support set to maximize the agreement prediction of the target pair.

of pairs,

$$\mathcal{H}_i^\star = \arg\max_{\mathcal{H}_i} \mathbf{S}(z_i, \mathcal{H}_i). \tag{2}$$

However, a key challenge arises in defining the similarity metric for pairs, $\mathbf{S}$. Existing methods (Radford et al., 2021; Li et al., 2022b;a) preliminary focus on aligning an image with its caption (Radford et al., 2021; Li et al., 2022a) from a image-text pair. They rarely emphasize on bringing similar pairs closer while distancing the dissimilar ones, which makes current methods fall short in gauging similarity between two pairs. For instance, the cosine similarity between two pairs is ill-defined, within the context of current methods.

**Selecting hard pairs by maximizing pair agreement.** To identify nearby pairs in the joint Vision-Language space, we introduce the idea of text-image pair agreement maximization. This can be viewed as a proxy task for selecting hard pairs. To illustrate why text-image pair agreement serves as an effective proxy for hard pair selection, we begin with the assumption that for a machine learning model, the similarity between the test sample and training data is a crucial factor that affects its prediction on the test sample. This assumption is supported by recent empirical and theoretical studies about model memorization (Chen et al., 2009; Zhang et al., 2021; Stephenson et al., 2021; Brown et al., 2021). Intuitively, if a pair agreement prediction model, trained on a set of pairs, predicts a particular target pair with a high probability of being a matching pair, then the target pair is likely similar to the matching pairs the model was trained on. With this in mind, the challenge of selecting hard pairs is reshaped into an optimization task centered on the text-image pair agreement, formally represented as:

$$\arg\max_{\mathcal{H}_i} \mathbf{S}(z_i, \mathcal{H}_i) = \arg\max_{\mathcal{H}_i} P_{\mathcal{M}}(z_i|\mathcal{H}_i), \tag{3}$$

where $P_{\mathcal{M}}(z_i|\mathcal{H}_i)$ denotes the prediction of a pair agreement model, $\mathcal{M}$, for the pair $z_i$ based on a pair set $\mathcal{H}_i$. This set is a subset of $\mathcal{D}_i$. In this framework, the goal of selecting hard pair is transformed into identifying a training set $\mathcal{H}_i$ such that the model $\mathcal{M}$ predicting the target pair as a matching pair.

Designing a suitable pair agreement prediction model for this proxy task is a nontrivial endeavor because the model needs to not only predict the pair matching probability but also allow the optimization of the training set, as indicated in Equation 3. Consequently, a conventional deep neural network design becomes unviable due to the impracticality of retraining across all possible sets $\mathcal{H}_i$ from $\mathcal{D}_i$. Taking inspiration from recent work (Norelli et al., 2022), we propose a data-centric design for the agreement prediction model $\mathcal{M}$. As illustrated in Figure 1, the model leverages two pretrained single-modal encoders, i.e., $f_{image}$ and $f_{text}$, to align representations of images and texts in a unified Vision-Language space. Specifically, the model encodes the target pair $z_i$ into $(I_i, T_i)$ using these single-modal encoders. For the visual modality, we

determine a similarity vector between the target pair $z_i$ and the dataset $\mathcal{D}_i$. The similarity vector is defined as $\vec{S}^I(x_i^I, \mathcal{D}_i) = [\dots, sim(I_i, I_j), \dots]^\top \in \mathbb{R}^{N-1}$. Here $I_j = f_{image}(x_j^I)$ with $(x_j^I, x_j^T)$ being an element of $\mathcal{D}_i$, and function $sim(\cdot, \cdot)$ denotes the cosine similarity. To counteract noise, values in the vector $\vec{S}^I(x_i^I, \mathcal{D}_i)$ are set to zero if $sim(I_i, I_j) < \tau$. This cleaned-up vector is represented as $\widetilde{S}^I$. The procedure for the textual modality is analogous, producing a vector denoted as $\widetilde{S}^T$. Note, the representations in this shared space are intuitively interpretable: each dimension corresponding to the visual/textual similarity of the input to a unique pair in the multimodal dataset. This interpretable characteristic enables us to directly optimize the supporting set to maximize the pair matching probability:

$$\mathcal{H}_i^\star = Argmax_{|\mathcal{H}_i|=k} \widetilde{S}^I(x_i^I, \mathcal{H}_i)^\top \widetilde{S}^T(x_i^T, \mathcal{H}_i), \tag{4}$$

where the $\mathcal{H}_i^\star$ is the hard pair set and $k \in \mathbb{R}^+$ is the number of selected pairs which is much less than $|\mathcal{D}|$. The previous problem can be efficiently solved by greedily choosing dimensions that maximize the inner product. Due to the interpretable property, the selected dimensions correspond to the desired pairs.

**Mitigation of noisy data impact.** The prior method assumes the target pair $z_i$ to be a suitable matching pair. However, in inherently noisy datasets, such as web-crawled ones like LAION (Schuhmann et al., 2022), mismatched pairs might be present. The potential adverse effects of hard pairs, which can be generated by these mismatched pairs, necessitates a strategy for their identification and elimination. To counteract the detrimental effect of such negative hard pairs arising from mismatched pairs, we employ a pair removal strategy based on the availability of hard pairs. The strategy proceeds as follows: A target pair $z_i$ is deemed as unsuitable and thus removed, if there is a non-empty subset $\mathcal{H}_i^{sub} \subseteq \mathcal{H}_i^\star$ with $|\mathcal{H}_i^{sub}| > 0$, such that $\widetilde{S}^I(x_i^I, \mathcal{H}_i^{sub})^\top \widetilde{S}^T(x_i^T, \mathcal{H}_i^{sub}) = 0$. This equation indicates that the number of matching pairs affirming $z_i$ as a genuine matching pair is less than $k$. For a dataset $\mathcal{D} \setminus z_i$, if there doesn't exist a small subset with size $k$ to support $z_i$ is indeed a matching pair, it suggests that the target pair is an outlier, possibly resulting from a mismatch. Such outliers can degrade dataset quality, and thus they are removed to ensure the reliability of hard data.

**Fast hard pair mining (FastHPM).** It is instinctive to infer that for a dataset collecting from a single source, the number of intrinsic hard pairs, *i.e.*, those robust enough to enhance the learned representation, will proportionally increase with the size of the dataset originating from that source. Thus, intuitively, to identify $k$ (much less than $|\mathcal{D}|$) "qualified" hard pairs, part of the dataset $\mathcal{D}$ is enough. Therefore, we present the Fast Hard Pair Mining (FastHPM) approach, devised to circumvent the time complexity associated with hard pair mining across the entire dataset. FastHPM's objective can be formalized as follows:

$$\mathcal{H}_i^\star \approx Argmax_{|\mathcal{H}|=k} \widetilde{S}^I(x_i^I, \mathcal{H}_i)^\top \widetilde{S}^T(x_i^T, \mathcal{H}_i), \tag{5}$$

where $\mathcal{H}_i \subseteq \overline{\mathcal{D}}_i$ and $|\overline{\mathcal{D}}_i| = C$ is sampled uniformly from set $\mathcal{D}_i$. In this equation, it's noteworthy that the selection of value $C$ is solely based on the number of hard pairs $k$, instead of the size of $\mathcal{D}_i$. Consequently, this optimization reduces the time complexity of FastHPM to $\mathcal{O}(N)$. The detailed procedure of the hard pair mining algorithm is consolidated and presented in Appendix A.1.

### 3.3 HNML: hard negative margin loss

The image-text contrastive loss $\ell_{CLIP}$, as illustrated in the preliminary section, aligns the true image-text pairs. But it poses no constraints on the overall geometry among data pairs (Goel et al., 2022). After involving hard data into the finetuning stage, equally maximizing the distance for normal negative pairs and hard negative pairs is an undesired way to utilize the information provided by hard negative pairs. The intuition follows directly from Figure 2. In a desired representation space, the similarity between the positive and the hard negative, $\mathbf{S}_1$, should be greater than the similarity between the positive and those normal negatives, $\mathbf{S}_2, \mathbf{S}_3$. Therefore, to impose the additional geometric structure, we introduce the Hard Negative Margin Loss (HNML):

$$\ell_{margin} = \frac{1}{|B|} \sum_{j \in B} \max\left(0, sim(I_i, T_j) - \min_{j' \in \mathcal{H}_i^p} \{sim(I_i, T_{j'})\}\right), \tag{6}$$

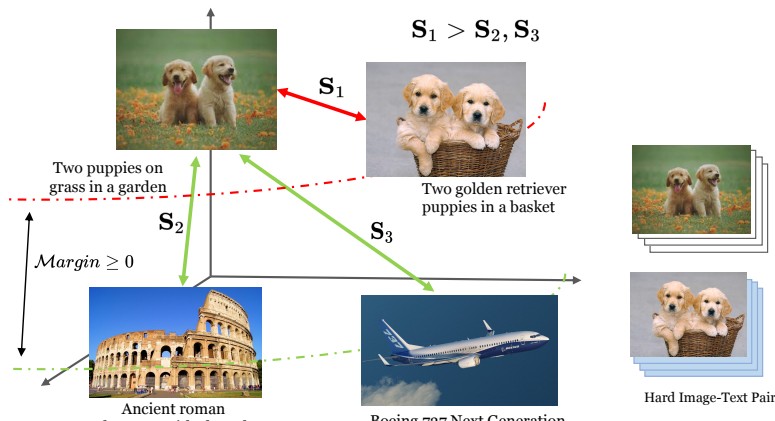

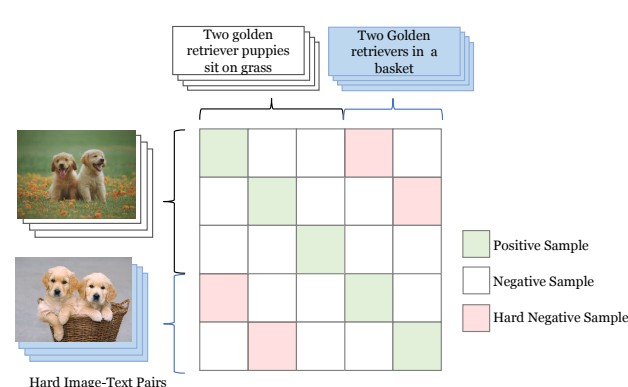

Figure 2: **Hard Negative Margin Loss (HNML).** Hard negative pairs are closer to the positive than the normal negative pairs.

Figure 3: **Further training CLIP with Hard Pairs.** For text-image pairs within a batch, we sample corresponding hard data from the preprocess hard pair set.

where $\mathcal{H}_i^p \subseteq \mathcal{H}_i^\star$ is the hard negative pairs for the target $z_i$ involved in one training batch. Note, the HNML is computationally efficient. No extra inner product computation is required. The geometric regularization is applied over the inner product matrix computed in the original CLIP loss, Equation equation 1. Then, the well-trained model is finetuned with the following loss, where $\gamma$ is the hyperparameter balancing the two losses,

$$\ell_{finetune} = \ell_{CLIP} + \gamma \ell_{margin}. \tag{7}$$

To boost the performance of well-trained CLIP models without introducing extra data and extra parameters, we introduce the further training strategy which involves the preprocessed hard pairs into the batch composition during training. As shown in Figure 3, for text-image pairs within the batch $B$, we randomly sample a subset $B'$ as seeds. Then, for $z_i \in B'$, we randomly select $|\mathcal{H}_i^p| = p$ pairs from $\mathcal{H}_i^\star$. The actual training batch is $\overline{B} = B \bigcup_{i=0}^{|B'|} \mathcal{H}_i^p$. We summarize the training pipeline in appendix A.1.

## 4 Experiments

In the experiments, detailed in Section 4.2, we showcase the potential of the HELIP approach to boost zero-shot classification, linear probing, and zero-shot image-text retrieval performance for vision-language models. In Section 4.3, the efficacy and robustness of HELIP in handling noisy datasets are explored. Section 4.4 offers comparative insights, highlighting the superiority of the Hard Pair Mining (HPM) method over other techniques that rely solely on information at the sample level. Lastly, in Section 4.5, we delve into the impact of the Hard Negative Margin Loss (HNML), emphasizing its role in maximizing the information derived from challenging pairs.

### 4.1 Experimental setup

**Training datasets.** In our experiments, we employed open-source datasets from multiple sources. These included the Conceptual Captions 3M (CC3M) (Sharma et al., 2018), Conceptual Captions 12M (CC12M) (Changpinyo et al., 2021). Additionally, we utilized two distinct 15M subsets of the YFCC100M (Thomee et al., 2016) dataset: the first, v1, collected by Radford et al. (2021), and the second, v2, collected by Li et al. (2022b). The combined datasets of CC3M, CC12M, and YFCC15M v1, which we denote as Open29M following the term used in previous work (Li et al., 2022b), were not completely obtained due to expired Urls. Additionally, we independently sampled 7.5M and 8M subsets from the noisier data source, LAION-5B (Schuhmann et al., 2022), labeled as LAION7.5M and LAION8M, respectively.

While these datasets are smaller than the 400 million pair dataset used in CLIP's original study (Radford et al., 2021), they are well-suited for the data and computational resources we have and have been frequently employed in benchmark evaluations for numerous studies on language-image pretraining (Goel et al., 2022; Li et al., 2022b; Mu et al., 2022).

**Downstream datasets.** We mainly verify the effectiveness of our methods through zero-shot image classification, linear probing, and zero-shot image-text retrieval. For zero-shot classification, in addition to commonly used ImageNet (Deng et al., 2009), CIFAR10, and CIFAR100 (Krizhevsky et al., 2009), we also verify the performance on 7 fine-grained classification datasets including Caltech101 (Fei-Fei et al., 2004), Food101 (Bossard et al., 2014), Sun397 (Xiao et al., 2010), Flowers102 (Nilsback & Zisserman, 2008), CUB (Wah et al., 2011), Stanford Cars (Krause et al., 2013) and FGVC Aircraft Maji et al. (2013). For the zero-shot image-text retrieval task, MS-COCO (Lin et al., 2014) and Flickr30K (Plummer et al., 2015) are adopted.

**Implementation details.** Our experiments are conducted across three distinct architectures: ResNet-50, ViT-B/16, and ViT-B/32, tailored to various datasets and pretrained models. Specifically, when pretraining on CC3M and CC12M, we utilize the ResNet-50 architecture with the CLIP model. For experiments involving the SLIP model on CC3M and CC12M, we employ ViT-B/16 to align with the framework established in Mu et al. (2022). In contrast, for pretraining on YFCC15M v1, v2, and Open29M datasets, we consistently use ViT-B/32 to ensure a fair comparison with the results reported in Li et al. (2022b). Furthermore, for the SLIP and DECLIP models, we adapt the pretrained parameters from the publicly available resources* The input resolution of the image encoder is $224 \times 224$ and the maximum context length of the text encoder is 77. All of our experiments are conducted on 8 V100 GPUs with a batch size of 128 for ViT-B/16 models, and a batch size of 512 for ResNet-50 models and ViT-B/32 models. The dimension of the image and text embeddings is 1024 for ResNet-50 models and 512 for ViT-B/16 and ViT-B/32 models. We set $\tau = 0.5$, $\gamma = 1$ and $p = 1$ for all the expriments by default. Automatic mixed-precision is used to save GPU memory. To avoid the model from overfitting to potential harmful distribution induced by the batch of hard data, we use early stopping if there's no performance gain within 5 consecutive epochs. Following the pre-training configurations established by Ilharco et al. (2021) and Mu et al. (2022), we set the training epoch at 32 for the baseline models that need to be pre-trained from scratch. Typically, best performance is achieved within these 32 epochs. The checkpoint demonstrating the highest performance is then chosen and saved for further use with HELIP. Unless specified, in the preparation of hard negative pairs, as suggested in the previous work Norelli et al. (2022), we employ the unsupervisedly pretrained vision transformer, DINO VITs8, as the image encoder (Caron et al., 2021). As for the text encoder, we utilize the SentenceT (Reimers & Gurevych, 2019), which is a transformer trained on a dataset comprising over 1 billion sentences gathered from the internet. To reflect that our method is designed to work with minimal assumptions regarding the encoders, we utilized encoders pretrained over single-modal source, instead of multimodality pretrained ones. Specifically, we employed an unsupervisedly pretrained vision transformer, DINO VITs8 (Caron et al., 2021), and a Sentence Transformer (SentenceT) (Reimers & Gurevych, 2019) as our text encoder. The embedding sizes are 384 for DINO VITs8 and 768 for SentenceT. For more details, we refer the readers to appendix.

## 4.2 Main results and discussion

**Zero-shot classification.** We compare zero-shot performances of the CLIP, SLIP, DECLIP, and those models finetuned by HELIP on CC3M, CC12M, YFCC15M and Open29M. We denote the models finetuned by HELIP as CLIP-HELIP , SLIP-HELIP , and DECLIP-HELIP respectively. As shown in Table 1, models further finetuned by HELIP consistently get significant improvements on the three datasets compared with their counterparts. Specifically, on CC3M, with the help of HELIP , zero-shot classification accuracy on ImageNet of CLIP model was improved from 19.04% to 19.86%. While SLIP model has a performance boost of over 13%, compared with its original number, achieving 26.05% accuracy on ImageNet. We additionally include two baseline methods: CYCLIP (Goel et al., 2022) and CLOOB (Fürst et al., 2021) for reference. As for pretraining on CC12M, we directly adopted the checkpoints released by SLIP (Mu et al., 2022). SLIP-HELIP outperforms its counterpart by 4.47% in zero-shot accuracy on ImageNet. Due to the absence of openly accessible parameters for DECLIP on the CC3M and CC12M datasets, our analysis focused on

---

*\*https://github.com/facebookresearch/SLIP, https://github.com/Sense-GVT/DeCLIP.*

comparing DECLIP with DECLIP-Helip over the YFCC15M v2 dataset. In this context, we present the performance of the SLIP and DECLIP models, as pretrained and released by Li et al. (2022b), using their designated evaluation pipeline (denoted with ∗). Additionally, we have included the corresponding results derived from our own evaluation pipeline for a fair comparison. Notably, both SLIP and DECLIP showed improvements with Helip, averaging increases of 15.49% and 6.74%, respectively. Further, to demonstrate Helip's sustained efficacy across larger datasets, we assessed CLIP and CLIP-Helip on Open29M. The best performance of CLIP on Open29M was recorded at the 18th epoch, achieving a 42.32% zero-shot performance on ImageNet. Extending the training by two more epochs resulted in a slight decrease in performance, dropping to 42.25%. Remarkably, our HELIP method was able to enhance CLIP's performance from 42.32% to 46.33% with just *one additional training epoch.*

|  | Method | ImageNet | CIFAR10 | CIFAR100 |
|---|---|---|---|---|
| **CC3M** | CYCLIP (Goel et al., 2022) | 22.08 | 51.45 | 23.15 |
|  | CLOOB (Fürst et al., 2021) | 23.97 | - | - |
|  | CLIP† (Radford et al., 2021) | 19.04 | 33.06 | 13.77 |
|  | CLIP†-Helip | 19.86 | 34.05 | 14.13 |
|  | SLIP (Mu et al., 2022) | 23.00 | 65.61 | 34.69 |
|  | SLIP-Helip | **26.05** | **68.18** | **37.77** |
| **CC12M** | CLIP† (Radford et al., 2021) | 30.27 | 51.07 | 21.94 |
|  | CLIP†-Helip | 32.05 | 52.27 | 24.51 |
|  | SLIP (Mu et al., 2022) | 41.17 | 81.30 | 53.68 |
|  | SLIP-Helip | **45.64** | **82.31** | **53.79** |
| **YFCC15M** | SLIP (Mu et al., 2022) | 25.29 (34.30∗) | 60.19 | 26.80 |
|  | SLIP-Helip | 35.43 | 75.49 | 47.84 |
|  | DECLIP (Li et al., 2022b) | 36.05 (43.20∗) | 78.12 | 50.60 |
|  | DECLIP-Helip | **43.80** | **84.88** | **56.31** |
| **29M** | CLIP† (Radford et al., 2021) | 42.32 | 71.98 | 42.73 |
|  | CLIP†-Helip | **46.33** | **77.97** | **48.33** |

Table 1: **Zero-shot performance on ImageNet, CIFAR10 and CIFAR100.** The † indicates baselines pre-trained by us. For all other baselines, publicly available pre-trained parameters were used. Specifically for SLIP and DECLIP on YFCC15M, we report results from two sources: our evaluation using OpenCLIP's framework with pre-trained parameters released by Li et al. (2022b), and the performance originally reported in Li et al. (2022b), marked with ∗.

**Zero-shot fine-grained classification.** By leveraging challenging image-text pairs in contrastive learning, Helip amplifies the discriminative capability of the CLIP model's visual embedding. This improvement proves valuable in classification, particularly for fine-grained datasets. Our evaluation on 7 fine-grained classification datasets (Table 2) reveals that SLIP-Helip boosts the zero-shot accuracy of CC3M and CC12M pretrained SLIP on Caltech101 by 12.88% and 3.95% respectively. Both CLIP and SLIP models witness consistent improvements with their Helip counterparts.

**Linear probing.** The linear probing task trains a randomly initialized linear classifier on the feature extracted from the frozen image encoder on the downstream dataset. To accomplish this, we train the logistic regression classifier using scikit-learn's L-BFGS implementation (Pedregosa et al., 2011), with maximum 1,000 iterations on those 7 datasets. For each dataset, we search for the best regularization strength factor on the validation set over 45 logarithmically spaced steps within the range 1e-6 to 1e+5. Experimental results in Table 3 demonstrate that both CLIP-Helip and SLIP-Helip have consistent improvements over their counterparts on almost all 7 datasets. Note that on CC12M SLIP-Helip performs marginally better on 5 out of 7 datasets. It's probably because the self-supervision of SLIP (Mu et al., 2022) within the visual modal can be beneficial for learning fine-grained visual embedding, while SLIP-Helip doesn't include image self-supervision during the training. In addition, we did not match the training batch size as SLIP (Mu

| Dataset | Method | Caltech101 | Food101 | Sun397 | Flowers102 | CUB | Stanford Cars | FGVC Aircraft | Average |
|---------|--------|-----------|---------|--------|-----------|-----|--------------|--------------|---------|
| CC3M | CLIP | 42.14 | 13.02 | 27.08 | 13.37 | 3.45 | 1.08 | 1.02 | 14.45 |
| | CLIP-Helip | 48.08 | 13.11 | 28.94 | 13.61 | 3.70 | 1.17 | 1.11 | 15.67 |
| | SLIP | 54.01 | 16.03 | 29.19 | 12.06 | 4.70 | **1.21** | **1.50** | 16.96 |
| | SLIP-Helip | **66.89** | **17.05** | **33.69** | **15.16** | **4.85** | 1.19 | 1.29 | **20.12** |
| CC12M | CLIP | 63.78 | 31.53 | 37.86 | 19.56 | 7.32 | 14.22 | 2.49 | 25.25 |
| | CLIP-Helip | 64.85 | 36.49 | 38.22 | 24.73 | 8.58 | 15.59 | 2.97 | 27.35 |
| | SLIP | 76.33 | 52.33 | 44.96 | **31.81** | 10.50 | 22.53 | 3.06 | 34.50 |
| | SLIP-Helip | **80.28** | **54.86** | **47.53** | 31.39 | **10.56** | **25.67** | **4.08** | **36.34** |

Table 2: **Zero-shot performance on fine-grained image classification.** On a variety of fine-grained image classification benchmarks, models finetuned by Helip achieves consistent performance gains compared to their original counterparts.

et al., 2022) because of resource limitations. A combination of Helip and image self-supervision and larger training batch size may be a potential direction for achieving better linear probe performance.

| Dataset | Method | Caltech101 | Food101 | Sun397 | Flowers102 | CUB | Stanford Cars | FGVC Aircraft | Avg. |
|---------|--------|-----------|---------|--------|-----------|-----|--------------|--------------|------|
| CC3M | CYCLIP | 80.88 | 54.95 | - | 83.74 | - | 22.72 | 28.02 | - |
| | CLIP | 80.11 | 53.82 | 56.40 | 84.07 | 40.30 | 22.70 | 35.61 | 53.29 |
| | CLIP-Helip | 82.49 | 59.79 | 59.56 | 87.84 | 46.19 | 30.01 | 42.48 | 58.34 |
| | SLIP | 87.96 | 72.50 | 66.96 | 91.91 | 49.77 | 39.25 | 45.87 | 64.89 |
| | SLIP-Helip | **89.64** | **73.09** | **67.67** | **93.02** | **53.16** | **42.44** | **48.66** | **66.81** |
| CC12M | CLIP | 85.35 | 68.00 | 64.45 | 87.88 | 48.75 | 57.80 | 40.32 | 64.65 |
| | CLIP-Helip | 85.87 | 68.89 | 64.95 | 88.36 | 49.41 | 58.55 | 40.17 | 65.17 |
| | SLIP | **92.89** | 83.63 | 74.34 | 94.87 | **60.99** | 73.43 | 52.23 | 76.05 |
| | SLIP-Helip | 92.85 | **84.25** | **74.74** | **95.09** | 60.53 | **74.23** | **52.36** | **76.29** |

Table 3: **Linear probe performance on Fine-grained Image Classification.** We report the linear probe performance on a variety of classification benchmarks. On average, both the CLIP and SLIP pretrained on CC3M and CC12M are improved.

| Pretraining Dataset | Method | COCO | | Flickr30K | |
|---------|--------|---------|---------|---------|---------|
| | | R@1 ↑ | R@5 ↑ | R@1 ↑ | R@5 ↑ |
| CC3M | CLIP | 14.4 | 34.1 | 31.7 | 56.0 |
| | CLIP-Helip | 17.8 | 39.8 | 35.4 | 61.0 |
| | SLIP | 22.3 | 45.6 | 39.6 | 68.6 |
| | SLIP-Helip | **23.4** | **48.3** | **41.8** | **69.6** |
| CC12M | CLIP | 26.9 | 52.6 | 47.2 | 74.3 |
| | CLIP-Helip | 27.8 | 54.3 | 48.2 | 75.4 |
| | SLIP | 39.0 | 66.0 | 65.4 | **90.1** |
| | SLIP-Helip | **39.4** | **67.2** | **66.2** | 89.7 |

Table 4: **Zero-shot image-text retrieval results on MSCOCO and Flickr.** ↑ indicates higher is better. Combining with Helip, CLIP and SLIP show better performance.

**Zero-shot retrieval.** We evaluate Helip on zero-shot image-to-text retrieval tasks on MS-COCO (Lin et al., 2014) and Flickr30K (Plummer et al., 2015). We compare CLIP, SLIP, and their counterparts trained on CC3M and CC12M respectively in Table 4. As shown in the table, both CLIP and SLIP benefit from Helip.

### 4.3 Performance of HELIP on noisy dataset

We extend our investigation by analyzing the efficacy of the HELIP model on subsets of LAION7.5M and 8M, respectively, which are randomly sampled from LAION (Schuhmann et al., 2022). The encoders in the Hard Positive Mining (HPM) utilized pre-trained VITs8 and SentenceT, while the CLIP model was executed with ViT-B/16. The outcomes are compiled in Table 5. Upon analysis of the presented data in Table 5, it

|            | ImageNet | CIFAR10 | CIFAR100 | Caltech | Food | Sun  | Avg. |
|------------|----------|---------|----------|---------|------|------|------|
| CLIP-7.5M  | 23.5     | 34.6    | 14.5     | 58.9    | 28.6 | 25.3 | 30.8 |
| HELIP-7.5M | **25.8** | **39.9**| **16.7** | **61.9**| **34.1** | **28.2** | **34.4** |
| CLIP-8M    | 25.1     | 31.1    | 12.9     | 60.9    | 29.5 | **27.5** | 31.2 |
| HELIP-8M   | **26.5** | **38.8**| **14.6** | **62.3**| **33.1** | 26.6 | **33.7** |

Table 5: **Zero-shot performance of HELIP on two LAION subsets.** HELIP counterparts witness consistent improvements over almost all the datasets.

can be discerned that the HELIP model, in general, outperforms the CLIP model on both subsets for the majority of the datasets used for evaluation, namely ImageNet, CIFAR10, CIFAR100, Caltech, and Food. The average performance of HELIP is also higher in both subsets. In the 7.5M subset, HELIP yields a better performance across all datasets, including Sun, with an average performance improvement of 3.6%. For the 8M subset, while the CLIP model scores slightly higher on the Sun dataset, the overall performance of HELIP remains superior, with an average performance increase of 2.5%. These results, therefore, underscore the improved performance yielded by the HELIP model in comparison to the CLIP model over a noisy data source, showing its potential of boosting larger scale pretraining that involves noisier data.

### 4.4 Comparison with other hard data selection method

We evaluate the efficacy of the proposed method in enhancing the discriminative capacity of learned representations by comparing its zero-shot classification performance with that of other hard data mining strategies. As described in the Section 2, a common way to define hard data is through intra-modality similarity. Hence, we introduce the hard data mining methods depending on image similarity and text similarity and denote them as IM and TM correspondingly. For a given target pair, we compute the cosine similarity between its image/text representation and those of the remaining dataset. The image and text representations are encoded using a pretrained Resnet50 and BERT, respectively. As a global preprocessing step, both IM and TM methods mine hard negatives. Subsequently, we integrate the mined hard negative pairs into the training pipeline of the CLIP+IM and CLIP+TM methods and optimize the original contrastive loss to finetune the model. Additionally, we employ the hard negative contrastive loss, HN-NCE, proposed by Radenovic et al. (2023), as a baseline. HN-NCE up-samples the weight of hard-negatives identified by the current model. As shown in Table 6, when the CC3M pretrained CLIP model is combined with HELIP, the performance of our pair-level hard data mining method significantly outperforms other sample-level techniques. We visualize the mined hard data obtained from three different preprocessing methods, namely, hard pair mining (HPM), image similarity mining (IM), and text similarity mining (TM), in Figure 4. The image-text pairs selected by HPM are displayed in the first row, while the second and third rows show the pairs selected by IM and TM, respectively. We observe that the captions of the hard pairs mined with image similarity are only loosely connected with the image of the target pair. For samples mined by TM, their images are even mismatched with the caption of the target pair. The fact that pairs mined by TM is easier than IM is also reflected in the Table 6, where the zero-shot performance of the CLIP+IM method consistently outperforms the CLIP+TM method across three datasets.

### 4.5 Impact of hard negative margin loss

We investigate the impact of using hard negative margin loss (HNML) on the performance of the SLIP model. In particular, our attention is directed towards an analysis of the SLIP model's performance, which has been previously pre-trained on the CC3M dataset, when it is both further trained with HPM+HNML

|              | Imagenet | CIFAR10 | CIFAR100 |
|--------------|----------|---------|----------|
| CLIP + HELIP | **19.86** | **34.05** | **14.13** |
| CLIP + TM    | 16.70    | 28.71   | 9.67     |
| CLIP + IM    | 16.93    | 29.22   | 10.42    |
| CLIP + HN-NCE | 19.47   | 29.88   | 11.83    |

Table 6: **Comparison of zero-shot performance for CC3M-finetuned CLIP with hard pairs (by HELIP) and hard samples (by other methods).** HELIP shows superior performance, consistently outperforming local/global hard sample mining techniques by a significant margin.

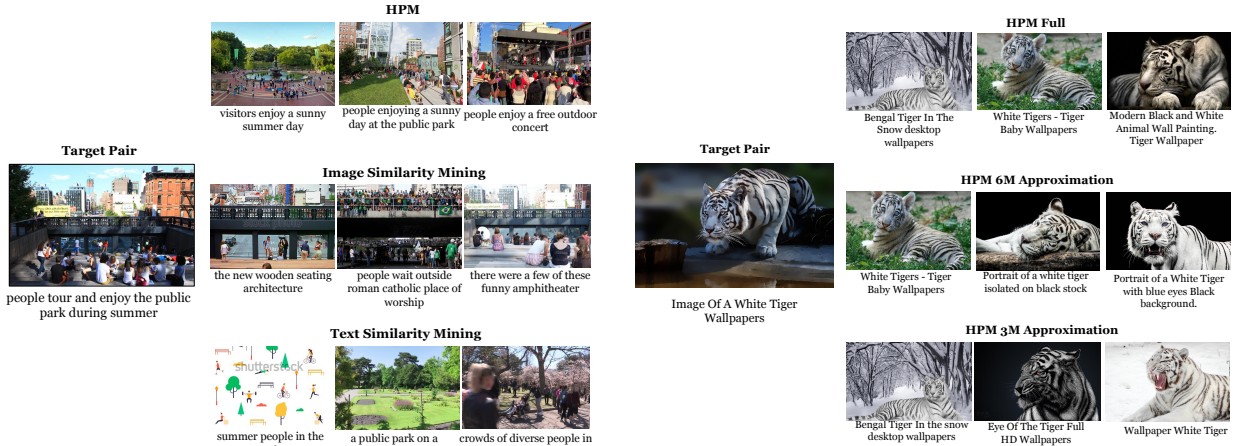

Figure 4: **Hard negative data selected by different methods.** Compared to text-image data mined using the visual or textual modality, hard pairs mined by HPM are more difficult.

Figure 5: **HPM and fastHPM.** We show the hard pairs mined by HPM and fastHPM. The quality of hard pairs mined by fastHPM is competitive with the pairs mined by HPM.

and left without HNML. Our approach involves a comparative analysis of the model's zero-shot classification performance across multiple datasets including ImageNet, CIFAR 100, CIFAR 10, Caltech 101, Food 101, and Sun397. The results of our evaluation are comprehensively detailed in Table 7. These demonstrate that the SLIP model supplemented with HPM and HNML exhibits superior performance, with a performance boost of 4.51 and 3.27 compared to the SLIP and SLIP + HPM models respectively. Interestingly, the model achieved superior performance on the CIFAR 10 dataset without HNML. We postulate that this may be attributed to HNML's ability to enhance the discriminative power of the learnt representations by employing the class distance as a cost metric. In light of this, our findings suggest that for classification datasets consisting of a larger number of subclasses, employing HNML during the training phase can lead to an increase in classification performance.

|         | ImageNet | CIFAR10 | CIFAR100 | Caltech101 | Food101 | Sun397 | Avg. |
|---------|----------|---------|----------|------------|---------|--------|------|
| SLIP    | 23.00    | 65.61   | 34.69    | 54.01      | 16.03   | 29.20  | 37.09 |
| wo HNML | 24.94    | **69.44** | 36.35  | 64.07      | 16.51   | 30.91  | 40.37 |
| w HNML  | **26.05** | 68.18  | **37.77** | **66.89**  | **17.05** | **33.68** | **41.60** |

Table 7: **SLIP finetuned with and without hard negative margin loss.** When finetuned with hard pairs, the zero-shot performance of CC3M pretrained SLIP can be further enhanced usingx HMNL.

|  | ImageNet | CIFAR10 | CIFAR100 | Avg. |
|---|---|---|---|---|
| CLIP Encoders | 19.57 | 33.28 | 13.53 | 22.12 |
| VITs8 + SentenceT | 19.86 | 34.05 | 14.13 | 22.68 |
| VITb16 + SentenceT | 19.62 | 35.53 | 14.67 | 23.27 |
| VITs8 + T5 | 19.61 | 33.99 | 13.82 | 22.47 |

Table 8: **The zero-shot performances of HELIP with different encoders in HPM.** HPM's performance is insensitive to the selection of encoders.

## 4.6 Delving into hard pair mining

**Impact of different encoders in HPM.** We explored the effect of different pretrained encoders on HPM's performance by alternating image and text encoders. Initially, the unsupervised pretrained DINO VITs8 (Caron et al., 2021) was paired with the SentenceT (Reimers & Gurevych, 2019) transformer, trained on over a billion internet-based sentences. This combination was later swapped for the SWAG VITb16 (Singh et al., 2022) and the T5 (Raffel et al., 2020). Additionally, experiments using OpenAI's CLIP model (Radford et al., 2021) multimodal encoders were conducted. Interestingly, as Table 8 suggests, the encoder choice seemingly has negligible impact on HPM's performance, likely due to the proficiency of current pretrained models in modeling intra-modal similarities. Moreover, the ability to use single-modal pretrained models and still achieve competitive or superior performance implies that there's no assumption of having access to a high-quality CLIP model, such as OpenAI's CLIP-400M.

**Performance Comparison between HPM and FastHPM.** A comparison was made between the zero-shot performances of SLIP models, further trained with hard pairs obtained from both HPM and fastHPM. This comparison, summarized in Table 9, was conducted under three different settings, each maintaining the hyperparameter $k = 500$. Additionally, we established subsets $\widetilde{\mathcal{D}}_i$ of sizes 3M and 6M, and accordingly denoted HELIP with these subset sizes as HELIP-3M and HELIP-6M. Table 9 shows that the zero-shot performances of HELIP-3M and HELIP-6M remain competitive with the global HPM hard pair mining approach. These findings suggest that fastHPM offers an efficient strategy for hard pair mining, without compromising performance. Additionally, they hint at fastHPM's potential to scale up hard pair mining in larger pre-training datasets, a promising direction for future exploration.

|  | Imagenet | CIFAR10 | CIFAR100 |
|---|---|---|---|
| SLIP | 41.17 | 81.30 | 53.68 |
| HELIP- 3M | 45.07 | **82.42** | 55.22 |
| HELIP- 6M | 44.98 | 81.64 | **56.62** |
| HELIP- Full | **45.64** | 82.31 | 53.79 |

Table 9: **Zero-shot performance for SLIP + Helip on CC12M with hard samples mined with HPM and fastHPM.** Compared with hard samples mined with HPM, the fast versions are competitive with the full version.

**Visual insights into HPM and FastHPM.** We took the initiative to visualize the hard pairs as identified by the aforementioned three methods. Within Figure 5, the leftmost image-text pairing is earmarked as the target. The pairs in the primary row are those selected via HPM. The subsequent rows, specifically the second and third, present image-text pairings identified by the 6M fastHPM and the 3M fastHPM methods, respectively. Through a comparative visualization, it's evident that the hard pairs pinpointed by fastHPM bear a significant resemblance to the target pair. For readers keen on delving deeper, we've provided an extended set of visualization outcomes in Appendix A.2.

**Computational time analysis.** Table 10 provides a comparison of the computational time required by HPM and fastHPM. The hard negative pairs preparation times listed were measured on 8 V100 GPUs, with the exception of the $*$ symbol, which was measured on a single V100 GPU. Given its efficiency and the

performance similarities observed in Table 9, fastHPM emerges as a compelling alternative to the full HPM method.

|            | CC3M    | CC12M   | YFCC15M |
|------------|---------|---------|---------|
| HELIP- 3M  | -       | 2h18min | 3h27min |
| HELIP- 6M  | -       | 5h3min  | 6h19min |
| HELIP- Full | 1h9min* | 9h11min | 17h41min |

Table 10: **Preparation time for hard pairs.** FastHPM speeds up the hard negative pairs mining process.

## 5 Conclusion

In this work, we explored the potential of boosting pre-trained CLIP models' performance by more effectively leveraging their original training dataset. This endeavor stemmed from recognizing the challenges posed by the loosely connected nature of web-crawled image-text pairs, which lead to suboptimal utilization of the available data by the standard CLIP loss. Our framework, HELIP, introduces a practical and efficient solution for boosting model performance without requiring extensive retraining or additional datasets. It utilizes challenging data from their original training datasets to improve the performance of contrastive learning. Specifically, HELIP defines hard negative data as the nearby pairs within the Vision-Language space. The Hard Pair Mining (HPM) module efficiently identifies challenging negative pairs, operating under the premise of access to single-modality pretrained models. Furthermore, the Hard Negative Margin Loss (HNML) effectively utilizes the information provided by those hard pairs during the fine-tuning phase. Our empirical results highlight the efficacy of the HELIP approach and emphasize the significance of pair-level hard data. This is demonstrated by notable improvements across various benchmarks such as zero-shot classification and image-text retrieval.

## 6 Future work

Moving forward, several avenues for future research present themselves. First, we aim to explore composition-aware fine-tuning for VLMs, which could potentially enable more effective utilization of multimodal information. Moreover, we are intrigued by the prospect of combining parameter-efficient tuning (He et al., 2022) with HELIP potentially further enhancing performance. Another area of interest is scaling up the dataset size and examining the applicability of the scaling law to our method. We also intend to investigate how the integration of our boosting algorithm might alter the multimodal dataset curation algorithm (Gadre et al., 2023). Ultimately, we hope our work will serve as a catalyst for additional research in the fine-tuning of pre-trained, large-scale multimodal models.

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

# A  Appendix

## A.1  Algorithm

We summarize the Hard Pair Mining (HPM), the fast Hard Pair Mining (fastHPM) and the training pipeline of HELIP in Algorithm 1, 2 and 3 respectively.

---

**Algorithm 1:** Hard Pair Mining (HPM)

---

**Input:** Hard pairs number per sample $k$
Pretrained unimodal vision model: $f_{text}$
Pretrained unimodal vision model: $f_{image}$
Dataset $\mathcal{D} = \{(x_1^I, x_1^T), (x_2^I, x_2^T), \cdots, (x_N^I, x_N^T)\}$
Threshold for visual and textual modality $\tau_I$ and $\tau_T$
**Output:** Hard samples $\mathcal{H} = [\mathcal{H}_1, \mathcal{H}_2, \cdots, \mathcal{H}_N]$
**for** $i \in [1, N]$ **do**
$\quad \mathbf{s} \leftarrow [0, 0, \cdots, 0]^\top \in \mathbb{R}^N$
$\quad I_i \leftarrow f_{image}(x_i^I)$
$\quad T_i \leftarrow f_{text}(x_i^T)$
$\quad$**for** $j \in [1, N]$ **do**
$\quad\quad I_j \leftarrow f_{image}(x_j^I)$
$\quad\quad T_j \leftarrow f_{text}(x_j^T)$
$\quad\quad \vec{S}_j^I \leftarrow \frac{I_i \cdot I_j}{\|I_i\|_2 \cdot \|I_j\|_2}$ **if** $\frac{I_i \cdot I_j}{\|I_i\|_2 \cdot \|I_j\|_2} > \tau_I$ **else** $0$
$\quad\quad \vec{S}_j^T \leftarrow \frac{T_i \cdot T_j}{\|I_i\|_2 \cdot \|T_j\|_2}$ **if** $\frac{T_i \cdot T_j}{\|T_i\|_2 \cdot \|T_j\|_2} > \tau_T$ **else** $0$
$\quad\quad \mathbf{s}_j \leftarrow \vec{S}_j^I \cdot \vec{S}_j^T$
$\quad$**end**
$\quad \mathcal{H}_i \leftarrow Argmax(\mathbf{s}, k)$
$\quad$**if** $\exists j \in \mathcal{H}_i,\ \mathbf{s}_j = 0$ **then**
$\quad\quad \mathcal{H}_i = \emptyset \quad$ # Indicate noise sample
**end**

---

Note, in the inner for loop, shown in Algorithm 1, the image and caption representations will be repeatedly computed. To accelerate the hard pair mining and avoid unnecessary computational overhead, we compute and save the encoded image features and text features. Besides, the outer loop is parallelized in the implementation.

## A.2  More visualization results

We offer further visualization results pertaining to the hard samples mined by various methods. As depicted in Figure 6, the hard samples sourced by HPM closely resemble the target sample (seen at the top left). Interestingly, for samples with fewer objectives, the image and text mining method can identify a reasonably challenging counterpart, as seen in the case of "the harbor in a small village". However, for intricate scenes, only the HPM is capable of yielding sufficiently challenging samples, like the scenario "people touring and enjoying the public park during summer". The dataset acquired from the web encompasses a myriad of such intricate cases. We posit that this is why training with hard samples unearthed by HPM yields more proficient outcomes.

Moreover, we present additional visualization results for hard samples mined via different techniques. Hard samples extracted by HPM exhibit a stronger resemblance to the target sample, as highlighted in Figure 6 (top left). We observed that the image and text mining methods can provide a relatively fitting hard counterpart for simpler samples, like "the harbor in a quaint settlement". However, for more intricate scenes, only the HPM method produces samples of adequate difficulty, such as "people touring and relishing the public

---

**Algorithm 2:** fast Hard Pair Mining (fastHPM)

---

**Input:** Hard pairs number per sample $k$
Pretrained unimodal vision model: $f_{text}$
Pretrained unimodal vision model: $f_{image}$
Dataset $\mathcal{D} = \{(x_1^I, x_1^T), (x_2^I, x_2^T), \cdots, (x_N^I, x_N^T)\}$
Threshold for visual and textual modality $\tau_I$ and $\tau_T$
Candidate pool size $C$
**Output:** Hard samples $\mathcal{H} = [\mathcal{H}_1, \mathcal{H}_2, \cdots, \mathcal{H}_N]$
**for** $i \in [1, N]$ **do**

    Uniformly $C$ samples from Dataset $\mathcal{D}$, $\overline{\mathcal{D}}_i = \{(x_1^I, x_1^T), (x_2^I, x_2^T), \cdots, (x_C^I, x_C^T)\}$
    $\mathbf{s} \leftarrow [0, 0, \cdots, 0]^\top \in \mathbb{R}^N$
    $I_i \leftarrow f_{image}(x_i^I)$
    $T_i \leftarrow f_{text}(x_i^T)$
    **for** $j \in [1, C]$ **do**
        $I_j \leftarrow f_{image}(x_j^I)$
        $T_j \leftarrow f_{text}(x_j^T)$
        $\vec{S}_j^I \leftarrow \frac{I_i \cdot I_j}{\|I_i\|_2 \cdot \|I_j\|_2}$ **if** $\frac{I_i \cdot I_j}{\|I_i\|_2 \cdot \|I_j\|_2} > \tau_I$ **else** $0$
        $\vec{S}_j^T \leftarrow \frac{T_i \cdot T_j}{\|I_i\|_2 \cdot \|T_j\|_2}$ **if** $\frac{T_i \cdot T_j}{\|T_i\|_2 \cdot \|T_j\|_2} > \tau_T$ **else** $0$
        $\mathbf{s}_j \leftarrow \vec{S}_j^I \cdot \vec{S}_j^T$
    **end**
    $\mathcal{H}_i \leftarrow Argmax(\mathbf{s}, k)$
    **if** $\exists j \in \mathcal{H}_i, \mathbf{s}_j = 0$ **then**
        $\mathcal{H}_i = \emptyset$     # Indicate noise sample
**end**

---

park throughout summer". The web-based dataset includes a significant proportion of these complex cases. Consequently, we infer that training with hard samples mined by HPM results in enhanced performance.

### A.3 Performance of HELIP with Scaled Training Data

In order to study the impact of expanding training dataset sizes on the effectiveness of HELIP, we trained CLIP on the YFCC15M, collected by Radford et al. (2021). This training resulted in a zero-shot classification accuracy of 25.46 on ImageNet. When applying HELIP, its performance reached 26.45 after one epoch. Summarizing the zero-shot performance on ImageNet of both standard CLIP and its enhanced counterpart, CLIP-HELIP, across varying data scales, we present those results in Figure 7. The results clearly demonstrate that HELIP consistently boosts CLIP's performance. Notably, the most significant improvement was observed with the Open29M dataset, where HELIP made an impressive performance increase of 3.06%. Extending this observation, we anticipate that HELIP will offer instant performance improvements for well-trained CLIP models on larger datasets, like the private 400M dataset referenced in Radford et al. (2021).

### A.4 Discussion about baselines

In our experiments, we utilized CLIP, SLIP, and DECLIP as baseline models on CC3M, CC12M, YFCC15M, and Open29M datasets. To ensure our results are both compelling and reproducible, we primarily employed publicly available checkpoints as our baseline and rigorously tested the effectiveness of HELIP against these checkpoints. On CC3M, the checkpoint of SLIP model is released[†]. We enhanced its performance by applying

---

[†]https://github.com/facebookresearch/SLIP#results-and-pre-trained-models

---

**Algorithm 3:** Hard samplE for boosting contrastive Language-Image Pretrained models (HELIP)

---

**Input:** $\mathcal{D} = \{(x_1^I, x_1^T), (x_1^I, x_1^T), \cdots, (x_N^I, x_N^T)\}$
Hard Pair Mining algorithm, HPM()     # or the fastHPM()
Pretrained unimodal vision model: $f_{text}$
Pretrained unimodal vision model: $f_{image}$
Pretrained contrastive language-image model $\{\phi_{image}, \phi_{text}\}$
hyperparameters:
  Hard pairs number $k$
  Hard negative margin strength $\gamma$
  Sampled hard negatives number $p$
  Learning ratio $\eta$
  Batch size $b$
  Training iteration number $E$
  Visual and textual modality threshold $\tau_I$ and $\tau_T$
**Output:** CLIP model $\{\phi_{image}, \phi_{text}\}$

$\mathcal{H} \leftarrow \text{HPM}(\mathcal{D}, f_{text}, f_{image}, k, \tau_I, \tau_T)$
**for** $iter \in [1, E]$ **do**

$\quad$ $B \leftarrow \{z_1, \ldots, z_b\} \overset{\text{i.i.d.}}{\sim} Uniform(\mathcal{D})$
$\quad$ **for** $z_i \in B$ **do**

$\quad\quad$ $\mathcal{H}_i^p \leftarrow \{z_i, \ldots, z_p\} \overset{\text{i.i.d.}}{\sim} Uniform(\mathcal{H}_i)$
$\quad\quad$ $\overline{B} \leftarrow B \cup \mathcal{H}_i^p$
$\quad$ **end**
$\quad$ Compute loss $\ell_{finetune}$, Equation (6), with samples $\overline{B}$ $\phi_{image} \leftarrow \phi_{image} + \eta \cdot \partial_{\phi_{image}} \ell_{finetune}$
$\quad$ $\phi_{text} \leftarrow \phi_{text} + \eta \cdot \partial_{\phi_{text}} \ell_{finetune}$
**end**

---

HELIP which notably improved the zero-shot performance on ImageNet from 23.00 to 26.05. However, we noticed that the CLIP with ResNet50 on CC3M is missing. To address this, we undertook the pretraining ourselves. Our results were encouraging: the performance of our pretrained CLIP with ResNet50 achieved a score of 19.86, surpassing the 17.10 achieved by SLIP's CLIP with ViT-B/32 as reported in Mu et al. (2022). This outcome suggests the robustness of our implementation. Besides, consistent with several prior studies, we found that on smaller pretraining datasets, CLIP with ResNet50 outperforms CLIP with ViT-B. On the CC12M dataset, a similar situation arose: while the SLIP checkpoint was available, the CLIP model was absent, leading us to undertake its pretraining. On the YFCC15M (v1) collected by Radford et al. (2021), we trained the CLIP model. This resulted in a 25.46 score in the ImageNet zero-shot classification, closely aligning with the 26.10 outcome reported by Cui et al. (2022). Additionally, for the YFCC15M (v2) dataset referenced in Li et al. (2022b), both SLIP and DECLIP pretrained parameters were made available by Li et al. (2022b), which we utilized directly as our baselines. On the larger dataset, Open29M, there was a lack of open-source pretrained checkpoints, prompting us to conduct the pretraining ourselves. Notably, the performance of our reimplementation (42.32) closely aligns with the results reported by Li et al. (2022b), indicating the effectiveness of our approach.

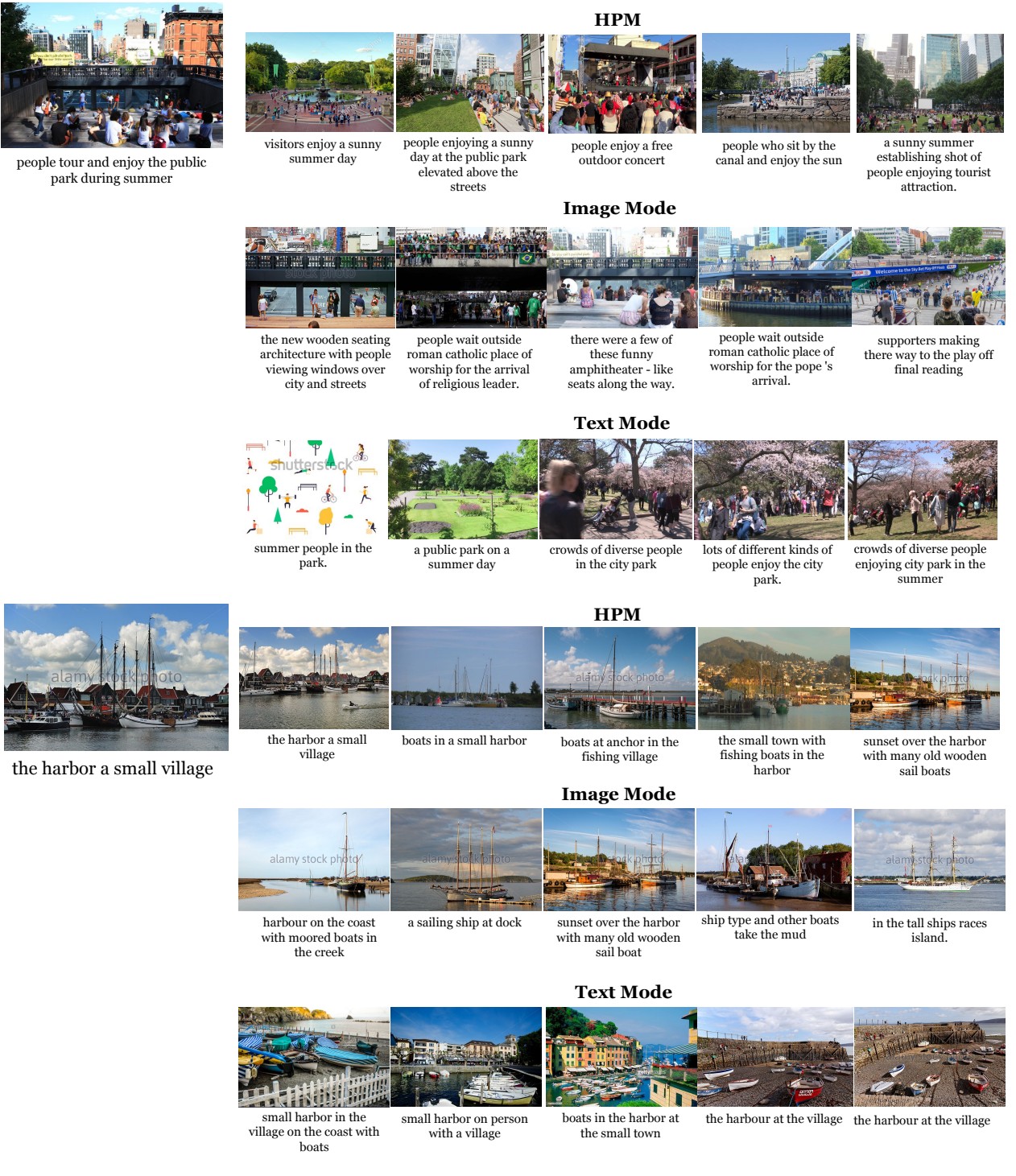

Figure 6: **Hard samples selected by different methods.**

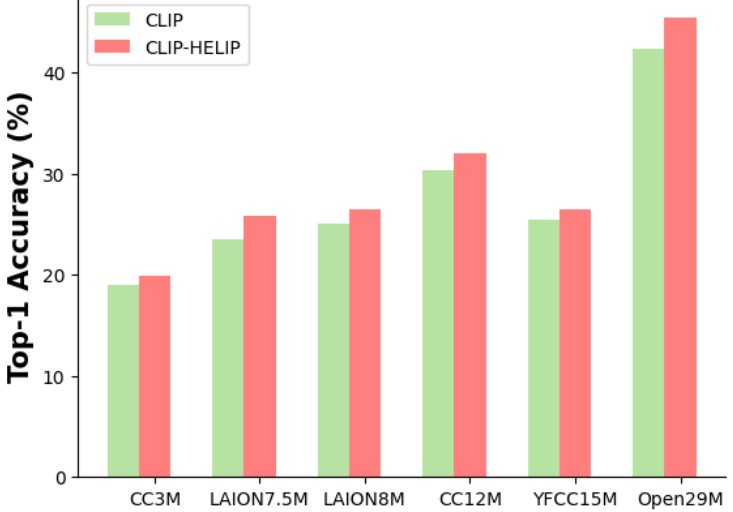

Figure 7: **Zero-shot performance on ImageNet for models trained over different size of dataset**.

