# OpenReview forum: "Boosting Visual-Language Models by Exploiting Hard Pairs"
_TMLR — Rejected by TMLR_

### Review · Reviewer_63LW · 2023-11-08

**Summary Of Contributions:**

This paper proposes a novel approach that finetunes the vision-language model on the pre-trained datasets by exploiting the hard pairs in the pre-trained datasets to boost the zero-shot performance of the model on the downstream benchmarks.

**Audience:**

Yes

**Claims And Evidence:**

Yes

**Requested Changes:**

1. The paper might need more fair comparisons between the experiments with and w/o the proposed approach
2. The paper needs clearer clarification of the motivation for doing another round of fine-tuning on the pre-trained dataset.

**Strengths And Weaknesses:**

Strengths: the overall improvement on various benchmarks shows that the proposed method, HELIP, is quite helpful.

Weakness: 1. However, the experiment comparison is not very convincing. The paper directly compares the pre-trained models and fine-tuned models. Although the training data are the same, it is still hard to identify if it is another round of training that achieves the improvements.
2. If the approach is so effective, why can't it be directly used in the pre-training? In other words, the motivation for applying the approach to the fine-tuning stage is unclear.
Others: The proposed approaches contain hard sample mining and margin loss, which are actually not very novel in the field of visual perception. However,  the simplicity of the proposed approach seems to be an advantage since it could be easier to be leveraged by other works.

Overall, my review is positive.

---

> ### Author Response · Authors · 2023-12-04
>
> We thank the reviewer for their encouraging feedback! We are pleased that they pointed out that "the proposed method, HELIP, is quite helpful on various benchmarks". We address all comments below.
>
>
>
> **Weakness 1 and Question 1: Need for clarification on fair comparisons (in terms of training epoch) between experiments with and without the proposed approach.**
>
> We agree with the reviewer's point regarding the need for clear comparisons, particularly in terms of training epochs, between experiments using our proposed approach and those without it. In training CLIP models, we adhered to the default number of training epochs recommended by OpenCLIP, which is 32. Typically, the checkpoint exhibiting the best performance is selected for saving. It's important to note that further training beyond this optimal checkpoint often leads to performance degradation. Applying our method allows for additional performance improvements even beyond this point. In order to address your concern, we extend the training epochs for baseline models. Specifically, we focused on CLIP training using the largest dataset in our experiment, Open29M. The best performance of CLIP on Open29M was recorded at the 18th epoch, achieving a 42.32 zero-shot performance on ImageNet. Extending the training by two more epochs resulted in a slight decrease in performance, dropping to 42.25. Remarkably, our HELIP method was able to enhance CLIP's performance from 42.32 to 46.33 with just one additional training epoch. This demonstrates the effectiveness of HELIP in a fair comparison.
>
> **Question 2: Clarification of the motivation for doing another round of fine-tuning on the pre-trained dataset.**
>
> Our work is driven by the goal of enhancing the performance of well-trained CLIP models without incurring significant costs, such as retraining the model or gathering additional text-image data. We identified a key challenge in the loosely connected nature of web-crawled image-text pairs, which often leads to less efficient data utilization by the standard CLIP loss. Our approach focuses on finetuning an existing model using hard pairs from the original dataset to achieve immediate performance improvements. We aim to provide a method that can effortlessly enhance an already trained model with minimal computational expense, making it more desirable and practical. Furthermore, to clarify our objectives, we have revised the Abstract and Conclusion sections of our paper.

---

> > ### Comment · Reviewer_63LW · 2023-12-05
> > **Official Comment by Reviewer 63LW**
> >
> > The responses make sense. Please include that "keep training CLIP leads performance degradation and adding a special fine-tuning step, especially HELIP can further improve the performance". I would like to change my score to a positive one.

---

> > > ### Author Response · Authors · 2023-12-14
> > >
> > > Thank you for your valuable feedback and for reconsidering your score based on the responses provided. We appreciate your suggestion and have updated the manuscript accordingly. Specifically, we have included the discussion in Sections 4.1 and 4.2. Thank you once again for your constructive input and positive reassessment!
> > >
> > >
> > > Best,
> > >
> > > Authors

---

### Review · Reviewer_sCbG · 2023-11-19

**Summary Of Contributions:**

The paper shows that contrastive self-supervised image-text pre-training (CLIP, SLIP) can be improved by using external models during hard negative mining. Authors propose to use this as an additional training step applied to already pre-trained models, but using the original data. They use it to select hard pairs from the entire corpus (rather than just from within a batch, as several other works are doing) and introduce Hard Negative Margin Loss (HNML) to modify the loss function within a batch to improve representation learning. Results show improvements on several zero-shot classification tasks, such as ImageNet, CIFAR as well as retrieval.

**Audience:**

Yes

**Claims And Evidence:**

Yes

**Requested Changes:**

I identified several critical issues that I would like to see addressed before publication:

- Please clean up some of the language: "plug-and-play" -- I don't know what that means; "pioneering plug-and-play", "revolutionary technique" -- I think it is too early to say that (and I am not sure tbh about the novelty of using external models to mine the corpus); "strategic approach to assimilate", "strategically utilizing the intrinsic data" -- again, what does that mean?
- Please clearly explain the significance and role of your baselines. The CLIP zero-shot performance on ImageNet is listed as 19.04% in Table 1, but the cited paper does not show that number. IIUC, this is a re-implementation of CLIP using different training data? Please show this in the Table.
- What happens if you scale up the training data, e.g. to all of the LAION data, instead of <10M sets? Do the benefits of hard mining diminish or disappear? It would be important to understand if and how the benefits of the proposed method translate to larger training datasets
- Please present the results of your work not in "chronological" but "logical" order. I.e. in Section 4.1 you explain that you use DINO ViTs8 and SentenceT as external image and text encoders (and explain why you pick those). In Table 8, you ablate and demonstrate that there does not seem to be much of a difference between which encoder is being used. It would seem more logical to start with CLIP (which matches the pre-training criterion), before ablating other methods?

**Strengths And Weaknesses:**

Strengths

The proposed framework is largely intuitive and elegant. It solves an interesting and timely problem, given that CLIP and similar methods are widely used for image, video and multimodal analysis. The paper is generally well written and accessible.

Weaknesses

The paper's language is often imprecise (see below) and the experimental results leave open several key questions. The most significant is that the CLIP training data is not available, so that the basic idea (better using the available training data) cannot be fully utilized, IIUC correctly. This should be explained better.

---

> ### Author Response · Authors · 2023-12-04
>
> We thank the reviewer for the thoughtful review and constructive comments! We are pleased that they pointed out our paper studies "an interesting and timely problem". We address all comments below, and we believe that the new, additional experiments (an experiment on a 29M pre-trained dataset and the corresponding analysis of our method with the increase in training set size) address the reviewer’s concerns.
>
>
> **Weakness 1 and Question 1: Jargon or phrases that are unclear and potentially cause overstatements.**
>
> We agree with the reviewer that some terms might cause confuse. For those terms and phrases mentioned by the reviewer, we have cleaned up those sentences. The revised sentences are highlighted in blue in the updated manuscript for your convenience.
>
>
>
> **Question 2: Clarification on the significance and accuracy of baselines.**
>
> This is a great suggestion! We further clarify the implementation of the baselines in Section 4.1 and added explanatory notes to the caption of Table 1. Furthermore, we offer a detailed discussion regarding the significance of the baselines, which we implemented, in the Appendix A.4.
>
>
>
> **Question 3: Impact of scaling up training data on the benefits of hard mining.**
>
> This is a great suggestion.  Investigating the effects of scaling up the training data on the efficacy of our proposed method is crucial for demonstrating its real-world value and applicability. We expanded our evaluation to include a larger (10x the size of CC3M) open-source dataset, Open29M (combining CC3M, CC12M, and YFCC15m). This extended analysis, detailed in sections 4.1 and 4.2 (highlighted in blue text), shows that the zero-shot performance of CLIP on ImageNet was boosted from 42.32 to 46.33 with only one finetuning epoch through the application of HELIP. These results demonstrate that HELIP can instantly enhance CLIP's performance on larger datasets.
> Moreover, we visualize the performance of models trained with and without HELIP in Appendix A.3. The results clearly demonstrate that HELIP consistently boosts CLIP's performance. Notably the most significant improvement was observed with the Open29M dataset, where HELIP made an impressive performance increase of 3.06\%. Extending this observation, we anticipate that HELIP will offer performance improvements for well-trained CLIP models on larger datasets.
>
>
> **Requested Change 4: Reorganization of results presentation from chronological to logical order.**
>
> Thanks for the suggestions. We have modified the section 4.1 and Table 8 in logical order in the updated revision.

---

> ### Author Response · Authors · 2023-12-14
>
> We appreciate the reviewer's thorough review of our paper. We have made every effort to address the concerns raised. If any explanations remain unclear, please let us know, and we will be happy to provide further clarification!
>
>
> Best regards,
>
> The Authors.

---

### Review · Reviewer_S2Q3 · 2023-11-20

**Summary Of Contributions:**

This article explores how to enhance the performance of pre-trained vision-language models by effectively utilizing hard pairs. Specifically, the article introduces a method for extracting hard text-image pairs through hard pair mining. Based on these hard pairs, the article proposes a hard negative margin loss to aid in the training of vision-language models. The effectiveness of this method is validated on multiple pre-trained models and various downstream tasks.

**Audience:**

Yes

**Broader Impact Concerns:**

NA.

**Claims And Evidence:**

Yes

**Requested Changes:**

See weaknesses.

**Strengths And Weaknesses:**

Strengths:

1. The topic addressed in this paper is significant. With the widespread application of pre-trained vision-language models in various downstream tasks, designing better pre-trained vision-language models is an important research area.
2. To the best of my knowledge, the use of hard pairs to enhance vision-language models has not been extensively studied, making this paper potentially the first of its kind.
3. In order to leverage hard pairs to improve vision-language models, the paper introduces a method called hard pair mining, which utilizes the similarity between the vision and language domains to identify these hard pairs. Additionally, a hard negative margin loss is proposed to complement the traditional contrastive loss.
4. The paper extensively validates the effectiveness of the proposed approach through experiments conducted on different pre-training datasets and various downstream tasks.


Weaknesses:

1. One major concern is related to the experimental validation of the paper. The pre-training datasets used in this study are relatively small, such as 'CC3M' and 'CC12M,' resulting in lower baseline results (e.g., approximately 30% accuracy on zero-shot ImageNet). In other words, the exploration in this paper is based on a relatively low baseline. However, current widely used vision-language models are based on larger datasets, such as LAION-2B, and achieve higher results (e.g., approximately 70% accuracy on zero-shot ImageNet). Therefore, it would be more valuable if the authors could validate the effectiveness of their method based on larger pre-training datasets and more practical baselines.
2. The writing could be improved. Some sentences are difficult to understand clearly, such as “Intuitively, pairs that most effectively support a model to make the decision that the target pair is matching are deemed its hard pairs“on Page Two.

---

> ### Author Response · Authors · 2023-12-04
>
> We thank the reviewer for the thoughtful review! We are glad that the reviewer pointed out "the topic addressed in this paper is significant", with a method that uses the hard pairs — an area yet to be explored and "extensively validates the effectiveness of the proposed approach through experiments".
>
> We addressed all of the comments below, and we believe that the new, additional experiments (an experiment on a 29M pre-trained dataset and the corresponding analysis of our method with the increase in training set size) resolve all of the reviewer’s concerns!
>
> **Weakness 1: Validation of the proposed method's effectiveness on larger datasets.**
>
> This is a great suggestion. Our initial focus was on enhancing the performance of CLIP, SLIP, and DECLIP on small pretraining datasets specifically CC3M, CC12M. This choice was driven by its alignment with our data and computational constraints, as well as its prevalent use in benchmarking language-image pretraining models[1,2,3]. Recognizing the need to test HELIP's scalability and effectiveness on larger datasets, we added expriment on larger dataset Open29M, a combination of CC3M, CC12M, and YFCC15M, which is about 10 times larger than CC3M.
>
>
>
> In Sections 4.1 and 4.2 of our revision (highlighted in blue for easy reference), we provide empirical result of HELIP 's impact on larger datasets. Notably, applying HELIP to CLIP resulted in a instant improvement in zero-shot performance on ImageNet, boosting the metric from 42.32 to 46.33 with only one epoch of fine-tuning. This outcome validates that HELIP can enhance CLIP's performance on larger datasets. Moreover, we visualize the performance of models trained with and without HELIP in Appendix A.3. The results clearly demonstrate that HELIP consistently boosts CLIP's performance. Notably the most significant improvement was observed with the Open29M dataset, where HELIP made an impressive performance increase of 3.06\%. Extending this observation, we anticipate that HELIP will offer instant performance improvements for well-trained CLIP models on larger datasets.
>
> **Weakness 2: Improvement of writing clarity for some sentences.**
>
> Thanks for your detailed review of the paper. We have revised the sentences you pointed out in the paper. The revised sentences are highlighted in blue in the updated manuscript for your convenience. Besides, we have carefully checked the entire paper and corrected grammatical errors and awkward phrasings.
>
>
> --
>
> [1] Goel, Shashank, et al. "Cyclip: Cyclic contrastive language-image pretraining." Advances in Neural Information Processing Systems 35 (2022): 6704-6719.
>
> [2] Li, Yangguang, et al. "Supervision exists everywhere: A data efficient contrastive language-image pre-training paradigm." arXiv preprint arXiv:2110.05208 (2021).
>
> [3] Mu, Norman, et al. "Slip: Self-supervision meets language-image pre-training." European Conference on Computer Vision. Cham: Springer Nature Switzerland, 2022.

---

> ### Author Response · Authors · 2023-12-14
>
> We thank the reviewer for the efforts in reviewing our paper. We tried our best to address the mentioned concerns. Are there unclear explanations here? We could further clarify them!
>
> Best,
>
> Authors.

---

### Author Response · Authors · 2023-12-04
**General Response**

We thank the reviewers for their thoughtful feedback! We are encouraged that they found "the topic addressed in this paper is significant" (R1), and the paper tackles an "interesting and timely problem" (R2). We are honored to hear that the reviewers found our methodology to be "intuitive and elegant" (R2), and that the exploration of hard pairs to enhance vision-language models "has not been extensively studied, making this paper potentially the first of its kind" (R1). Additionally, we are glad the reviewers highlighted that "the overall improvement on various benchmarks shows that the proposed method, HELIP, is quite helpful" (R3).

In response to the reviewers' feedback, we have refined our expressions and included new experiments, analyses, and discussions in the revision. For your convenience, these updates are highlighted in blue text:


1. We added an experiment on a larger ($\sim$ 10x the size of CC3M) open-source dataset, which we refer to as Open29M (combining CC3M, CC12M, and YFCC15M). This extended analysis, detailed in sections 4.1 and 4.2 (highlighted in blue text). The result indicates that HELIP can instantly enhance CLIP's performance on larger datasets.


2. We presented a discussion section about the effectiveness of HELIP with respect to scaled training data in Appendix A.3. The results show that HELIP consistently boosts CLIP’s performance across various training data sizes, with the most significant improvement observed in the Open29M dataset, which is the largest dataset employed in our experiments.

3. We further clarified the implementation and significance of the baselines. Besides, we provide more discussion about those baselines in Appendix A.4.

4. We refined and reorganized several sentences to enhance clarity and more effectively convey the motivation behind our work. The modifications made in the revision are highlighted in blue for easy identification.

Once again, we thank reviewers for their feedback and we're happy to answer any additional questions or discuss remaining concerns!

---

### Decision · Action_Editor_LzCT · 2024-01-03

**Recommendation:** Reject

**Comment:**

Although the paper shows significant improvements on some settings in different experiments, a common concerns of the reviewers is that there is not sufficient evidence to prove that the performance improvement come from the proposed loss. The rebuttal addresses some of the concerns of the reviewers, but some remained unsolved. For example, Reviewer sCbG pointed out that the proposed method relies on fully accessible to the pretrained data, what if pretrained data is not available? Furthermore, the current results are based on some small scaled data, so it is believed not a well-pretrained model. One suggestion would be what if we fine-tunes the original CLIP model with the proposed method and views it as a "domain adaptation" method? I.e. taking an 80% checkpoint and seeing if it improves by the proposed hard mining etc, even if we only use a small amount of data? Or use more data?

In conclusion, the paper is interesting and generally well written. It just has a lack of evidence to prove the usefulness of the proposed method. I would suggest the authors to revise the paper according to the reviewers' comments and resubmit. I am happy to supervise it in the next round.

**Audience:**

Audience in multi-modal pretraining could find the work interesting to read.

**Claims And Evidence:**

This paper proposes to incorporate hard pair mining into a new loss called hard negative margin loss, and show performance improvements on a variety of pretrained models. In general, the paper is smooth to read, and the descriptions are clear, however, some aspects of the proposed method are not well explained or tested (see more details below).

**Resubmission Of Major Revision:**

The authors may consider submitting a major revision at a later time.